# Retrospective Analysis of Patients with Signet Ring Subtype of Colorectal Cancer with Peritoneal Metastasis Treated with CRS & HIPEC

**DOI:** 10.3390/cancers12092536

**Published:** 2020-09-07

**Authors:** Aruna Prabhu, Andreas Brandl, Satoshi Wakama, Shouzou Sako, Haruaki Ishibashi, Akiyoshi Mizumoto, Nobuyuki Takao, Kousuke Noguchi, Shunsuke Motoi, Masumi Ichinose, Yang Liu, Yutaka Yonemura

**Affiliations:** 1Department of Surgical Oncology, Thangam Cancer Center, Namakkal 637001, Tamil Nadu, India; dranuprabhu.surgeon@gmail.com; 2Digestive Unit, Champalimaud Foundation, 1400-038 Lisbon, Portugal; andreas.brandl@fundacaochampalimaud.pt; 3Department of surgery, Graduate school of medicine, Kyoto University, Kyoto 606-8303, Japan; wakama9@kuhp.kyoto-u.ac.jp; 4Department of Regional Cancer therapy, Peritoneal Surface Malignancy Center, Kishiwada Tokushukai Hospital, Kishiwada, Osaka 596-0042, Japan; sako-rgb.009@cap.ocn.ne.jp (S.S.); haruaki.ishibashi@tokushukai.jp (H.I.); yang.liu@tokushukai.jp (Y.L.); 5NPO to Support Peritoneal Surface Malignancy Treatment, Japanese/Asian School of Peritoneal Surface Oncology, Kyoto 600-8189, Japan; 6Department of Regional Cancer Therapy, Peritoneal Surface Malignancy Center, Kusatsu General Hospital, Shiga 525-8585, Japan; mizumoto@kusatsu-gh.or.jp (A.M.); takao@kusatsu-gh.or.jp (N.T.); noguchi@kusatsu-gh.or.jp (K.N.); motoi@kusatsu-gh.or.jp (S.M.); ichinose@kusatsu-gh.or.jp (M.I.)

**Keywords:** colorectal cancer, signet ring sub-type, peritoneal metastasis, cytoreductive surgery, hyperthermic intraperitoneal chemotherapy

## Abstract

**Simple Summary:**

Approximately 1% of all patients with colorectal cancer, and 15% of patients with peritoneal metastasized colorectal cancer present with the subtype of signet ring cell, which is associated with inferior oncological outcome and reduced overall survival. The evidence whether patients with signet ring cell subtype are benefiting from cytoreductive surgery and hyperthermic intraperitoneal chemotherapy is limited. The aim of this large bicentric retrospective study including 60 patients with this subtype was to explore the survival and define predictive factors of these patients. Median overall survival was 14.4 months, while small bowel PCI > 2 (HR: 6.5; *p* = 0.008) was the strongest predictive factor for inferior patient survival. The study concludes, that after thoroughly selection patients for CRS and HIPEC, even patients with signet ring cell subtype of colorectal cancer may benefit from this concept.

**Abstract:**

Signet ring cell subtype (SRC) of colorectal cancer (CRC) is a rare subtype and occurs in approximately 1% of all patients with CRC. Patients with peritoneal metastasis (PM) of SRC have a poor prognosis, and this subtype is frequently considered as a contra-indication for extensive surgical treatment. This retrospective study from two dedicated peritoneal surface malignancy centers in Japan included all patients treated with CRS ± hyperthermic intraperitoneal chemotherapy (HIPEC) between July 1994 and December 2017 from a prospectively maintained database. Preoperative, operative, and postoperative parameters were recorded, including complication rates and follow-up. Sixty of the 320 patients treated with CRS due to CRC were diagnosed with SRC subtype. The mean age of the patients was 51.4 years, and the mean peritoneal carcinomatosis index (PCI) was 13.1. Complete cytoreduction was achieved in 61.7% of cases. The postoperative morbidity rate was 25% and the mortality rate was 1.7%. The median overall survival (OS) was 14.4 months. Cox regression analysis revealed small bowel PCI > 2 (hazard ratio (HR) 6.5; *p* = 0.008) as the most important factor for OS. With accurate patient selection (e.g., PCI ≤ 12 or small bowel PCI ≤ 2), even patients with PM of CRC with SRC subtype may benefit from CRS and HIPEC, with median OS from 17.8 to 20.8 months and 5-year OS of 11.6%.

## 1. Introduction

Peritoneal metastases (PMs) in colorectal cancer (CRC) are diagnosed in 4–10% of patients at the initial diagnosis. Additionally, 15–20% of patients develop peritoneal recurrent disease [1,2,3,4]. These patients are commonly treated with palliative chemotherapy with or without palliative surgery. Even with the use of newer chemotherapeutic drugs (such as oxaliplatin or irinotecan) or the use of targeted agents (such as bevacizumab or cetuximab), the survival of this group of patients with PM remains dismal. The reported median overall survival (OS) for patients treated with systemic chemotherapy with or without targeted therapy is in the range of 10–12 months for patients with PM compared to 17–22 months for patients without PM [5,6,7,8]. 

With the introduction of cytoreductive surgery (CRS) and hyperthermic intraperitoneal chemotherapy (HIPEC), there has been a significant improvement in median OS in selected patients. Elias et al. reported a median survival of 63 months in these patients when treated with CRS and HIPEC [9]. In a randomized control trial, Verwaal et al. was able to demonstrate a median survival of 22.2 months in patients treated with CRS and HIPEC compared to 12.6 months in patients treated with systemic chemotherapy [10]. Similarly, Franko et al., Quenet et al., Froysnes et al., and Baratti et al. showed improved survival in selected patients of CRC with PM after treatment with CRS and HIPEC [11,12,13,14]. 

Recently, the results of the randomized controlled trial Prodige 7 were published. This trial compared CRS alone versus CRS with HIPEC in combination with perioperative chemotherapy in patients with PM of CRC [15]. Both groups showed similar median OS: 41.2 and 41.7 months, respectively. This study further confirmed the improvement in survival with CRS, but the use of HIPEC did not show any additional benefit in survival. However, a subgroup of patients with peritoneal carcinomatosis index (PCI) of 11–15 did receive benefit with the addition of HIPEC. Further studies will help us in selecting the group of patients that would receive the maximum benefit from CRS and HIPEC. 

Thus, though the survival of patients with PM of CRC has shown significant improvement, the role of CRS and HIPEC for patients with signet ring subtype of CRC remains unclear. Many groups consider signet ring histology as a relative contra-indication to proceed with CRS and HIPEC. In a meta-analysis of 25 studies by Kwakman et al., signet ring histology was identified as a negative prognostic factor after CRS and HIPEC [16]. Signet ring is a rare subtype of colorectal cancer, accounting for approximately 0.9% to 1% of all colorectal cancer cases [17,18]. It is known to be associated with a poor prognosis, with most patients presenting in an advanced stage of disease [19,20]. Peritoneum is the most common site of metastasis, with reported 5-year OS between 0–3% in patients with peritoneal carcinomatosis [20,21,22]. Data on patients with signet ring subtype treated with CRS and HIPEC have reported no improvement in survival, with the reported median OS in these patients being 7–13 months even after treatment with CRS and HIPEC [23,24,25]. We hereby report a retrospective analysis of 60 cases of signet ring CRC with PM treated with CRS and HIPEC.

## 2. Results

From July 1994 to December 2017, 320 patients of CRC with PM underwent CRS and HIPEC. Out of these 320 patients, 60 patients had a pathology confirming signet ring subtype. Baseline demographic, clinico-pathological and treatment characteristics are shown in Table 1. In total, 34 patients were male and 26 were female. The mean age at the operation day was 51.4 years (SD: 13.9).

The sigmoid colon was the most common site of primary tumor origin (30%), followed by the ascending colon (23.3%). Synchronous PM occurred in 60.3% of all patients. The mean PCI was 13.1 (SD: 10.4) with a mean SB-PCI of 3.6 (SD: 3.8). Complete cytoreduction (CC0) was achieved in 61.7% of all patients and CC0/1 in 66.7%. HIPEC was performed after CRS in 45 patients. Fifteen patients did not undergo HIPEC based on intraoperative course (excessive blood loss, inability to maintain adequate urine output or mean arterial pressure). The post-operative morbidity was 25% and the post-operative mortality was 1.7%, with one patient dying in the post-operative period due to uncontrolled intra-abdominal hemorrhage. 

The mean follow-up was 15.2 months, and the median OS for patients treated with CRS and HIPEC was 14.4 months (Figure 1). The median progression-free survival (PFS) was 4.4 months (Figure 2). Notably, one patient with stomal site recurrence was operated upon and now is disease free. The 1-, 2-, 3-, and 5-year OS were 55.2%, 25.9%, 15.5%, and 11.6%, respectively. Patients treated with CRS and HIPEC between 1994 and 2013 showed inferior overall survival compared to patients treated between 2014 and 2017 (11.8 vs. 20.8 months; *p* = 0.02; Figure 3). 

### 2.1. Analysis of Predictive Factors 

Univariate log-rank test of survival data showed that male gender was associated with a worse survival as compared to females (11.8 vs. 24.4 months, *p* = 0.004). Patients who underwent a CC0/1 resection survived longer than with a CC2/3 resection (17.6 vs. 4.7 months, *p* < 0.001). Use of HIPEC in addition to CRS was associated with improved median OS as compared to those patients in whom HIPEC could not be used (17.3 vs. 5.5 months, *p* = 0.007). Also, patients with a negative intra-operative cytology had better median OS than those with a positive cytology (17.6 vs. 9.6 months, *p* = 0.017). Patients with a PCI score of ≤12 had better median OS as compared to patients with PCI > 12 (17.8 vs. 10 months, *p* = 0.004) and so did the patients with small bowel (SB)-PCI ≤ 2 as compared to patients with SB-PCI > 2 (20.8 vs. 3.5 months, *p* < 0.001). Age > 60 years, synchronous or metachronous appearance of peritoneal metastasis, side of primary tumor (right sided or left sided), and the nodal status (negative or positive lymph nodes) did not significantly influence the OS.

Thus, on univariate analysis (Table 2), female sex, negative intra-operative cytology, completeness of cytoreduction (CC0/1), HIPEC, PCI ≤ 12, and SB-PCI ≤ 2 were statistically significantly associated with improved median OS.

In the Cox regression model, SB-PCI > 2 and non-performance of HIPEC retained their statistical significance, with hazard ratio (HR) 6.51 (*p* = 0.008, 95% CI: 1.65–25.74) and HR 2.20 (*p* = 0.049, 95% CI: 1.01–4.81), respectively (Table 2). For PFS, however, only intraoperative cytology status was noted to have statistically significant influence, in that patients with negative intra-operative cytology had better PFS than those with positive intra-operative cytology (9.3 vs. 2.9 months, *p* = 0.038).

### 2.2. Tumor Recurrence

In patients who had undergone CC0/1 resection, 37 patients experienced recurrence (37/40). Isolated peritoneal recurrence was noted in 15 (37.5%) patients, and systemic recurrence without peritoneal recurrence was noted in 14 (35%) patients. Eight patients had both peritoneal as well as systemic recurrence (i.e., 20% of patients). In all of the 22 patients with systemic recurrence, liver metastasis occurred in 13 patients, lung metastasis in 5 patients, pleural metastasis in 1 patient, bone metastasis in 5 patients, lymph nodal metastasis in 5 patients (2 patients had extra-abdominal lymph nodal metastasis: 1 patient in the mediastinum and the other in the supra-clavicular region).

## 3. Discussion

Signet ring subtype is known to be a poor prognostic factor for patients with metastasized CRC. There are only a few published papers about the use of CRS and HIPEC in this sub-group of patients. Our study provides more evidence, and is the largest study to our knowledge, showing the benefit of CRS and HIPEC in selected patients. We were able to demonstrate a 5-year overall survival of 11.6%, which until now has not been reported. 

In 2004, Verwaal et al. reported survival of 102 patients with peritoneal metastasis from gastro-intestinal primary lesions that were treated with aggressive cytoreduction and HIPEC. They identified signet ring histological subtype to be associated with decreased OS (HR: 2.24, 95% CI: 1.21–4.16, *p* = 0.008) and reported a median OS of 13 months in the 15 patients with signet ring subtype [23]. However, they did mention that a few patients were long-term survivors and that treatment should not be denied solely based on histology.

Subsequently, Chua et al. reported on 15 patients with PM from CRC signet ring subtype that were treated with CRS and HIPEC. They reported a poorer median OS of 13 months for patients treated with CRS and HIPEC as compared to 18 months for patients treated with systemic chemotherapy alone (*p* = 0.75) [24]. However, in this study, more than 50% of the patients did not undergo complete CRS (8/15). It has been consistently shown across various studies that patients with CC0/1 resection receive the maximal survival benefit after CRS and HIPEC.

In 2012, van Sweringen et al. reported median survival of 7.2 months in the 11 patients with signet ring subtype treated with CRS and HIPEC [25]. The average PCI was 15. Patients with CC0/1 resection had a median survival of 13 months as compared to 6.4 months after CC2/3 resection. The patients included in this study had peritoneal carcinomatosis from various gastro-intestinal primaries, and only 5 patients (out of the 36) had a colonic primary. 

In 2014, Winer et al. reported on the survival of 30 patients with peritoneal carcinomatosis from colorectal origin signet ring subtype that were treated with CRS and HIPEC. The median OS was 12.2 months, with 5-year OS of 7% [26]. The average PCI was 11.5, and 77% of patients underwent CC0/1 resection. Patients with CC0/1 had a median OS of 15.8 months as compared to 2.4 months with CC2/3 resection (*p* = 0.011), thus demonstrating modest benefit in survival, after CC0/1 resection [26].

In another study by van Oudheusden et al., 20 out of the 351 patients of CRC-PM that were treated with CRS and HIPEC had signet ring histology. Median OS of the signet ring subtype was 14.1 months with 0–3-year OS. A macroscopically complete cytoreduction was achieved in 87.5% of patients [27]. All of these studies are illustrated in Table 3.

In our retrospective study, the median OS was 14.4 months with a 5-year OS of 11.6%. Patients treated in more recent years (2014–2017) showed a median OS of 20.8 months. Patients with CC0/1 resection, a negative intra-operative cytology, treated with HIPEC in addition to CRS and PCI of ≤12 had higher median OS of more than 17 months. Patients with SB-PCI ≤ 2 had a median OS of 20.8 months, which until now has never been reported. The small bowel PCI ≤ 2 was previously reported as a significant factor influencing the OS of patients treated with CRS and HIPEC due to CRC in a study by Elias et al. in 2014 [28].

An important aspect that may have influenced the outcome in our patients was the use of neoadjuvant chemotherapy (NACT) and therefore patient selection, as patients with tumor progression after NACT were excluded. In our retrospective study, 96.7% patients were treated with neoadjuvant systemic chemotherapy, with 24 patients receiving additional targeted therapy (anti-VEGF/anti-EGFR/both) and 3 patients receiving additional IP chemotherapy. Also, in the study by Winer et al., which did show some benefit of CRS and HIPEC in this subgroup of patients, 83.3% were treated with NACT [26]. The rate of NACT in the study reported by van Oudheusden et al. was only 21.4% [27]. Whether the use of neoadjuvant treatment may have downsized the metastatic lesions, and hence improved rates of complete cytoreduction and survival, is a matter of speculation. We are currently looking at tumor regression grades in these patients, which may help us answer this question definitively.

Though we have demonstrated a median OS of 14.4 months and 5-year OS of 11.6% in this otherwise poor prognostic subgroup, our study does have a few limitations. Firstly, it is a retrospective analysis. The number of patients, though greater than the numbers reported in other studies, is still small. Second, the inclusion period of patients is relatively long (23 years), and therefore treatment strategies and chemotherapeutic regimen (intravenous and HIPEC) varied between patients based on drug development and changing evidence. Data on tumor regression grading after use of neoadjuvant chemotherapy in these patients is awaited. Having said that, this is a relatively rare subtype, and it is difficult to conduct a prospective randomized controlled trial. In the future, with many centers collaborating, a prospective study may be planned to give us a higher level of evidence in the treatment of this subgroup of patients. Until then, just a histological subtype of signet ring should not deter us from offering CRS and HIPEC in those properly selected patients who could benefit from this treatment. 

## 4. Methods

This retrospective study was conducted on patients treated at two prominent hospitals in Japan, which have been treating patients with peritoneal surface malignancies for more than 30 years. Demographic, treatment-related, and follow-up details were obtained from the prospectively maintained database. 

In total, 320 patients with peritoneally metastasized CRC were treated with CRS and HIPEC between July 1994 and December 2017. Amongst these 320 patients, 60 patients were diagnosed with signet ring subtype of CRC.

All included patients had a histologically proven diagnosis of signet ring subtype of CRC. Patients with good performance status (Eastern Cooperative Oncology Group: ECOG 0/1) and with normal cardiac, renal, liver, and bone marrow functions were selected for undergoing this comprehensive treatment. Patients with extra-abdominal lymph nodal and/or hematogenous metastases were excluded. These patients were treated with palliative chemotherapy ±radiation along with best supportive care.

### 4.1. Treatment and Follow-Up Protocol 

A detailed written and informed consent was obtained from all patients, and the treatment was acknowledged by the ethical committee of Kishiwada Tokushukai Hospital (H19-3) and Kusatsu General Hospital (2012-1026-01). Patients were treated with NACT: FOLFOX (5-fluorouracil, leucovorin, oxaliplatin), FOLFIRI (5-fluorouracil, leucovorin, irinotecan), XELOX (Capecitabine, oxaliplatin), XELIRI (Capecitabine, irinotecan), IRIS (Irinotecan, S1: Tegafur/gimeracil/oteracil), and/or SOX (S1: Tegafur/gimeracil/oteracil, oxaliplatin). On average, patients were treated with 5–6 cycles of NACT. Ten patients received more than one regimen in the neoadjuvant setting. The choice of regimen depended on the following aspects: metastasis being synchronous or metachronous, treatments received in the past, disease-free interval, and patient’s tolerability. Some of the patients also received targeted agents, according to their tumor biology: anti-VEGF (vascular endothelial growth factor); bevacizumab and/or anti-EGFR (epidermal growth factor receptor); cetuximab or panitumumab, along with NACT. Three patients received neoadjuvant intraperitoneal (IP) chemotherapy, docetaxel + cisplatin, after IP port placement under local anesthesia. The technique of IP port placement and treatment protocols have already been described by our group [29]. Response evaluation was done with CT scan, and those with response or stable disease underwent CRS + HIPEC.

Cytoreductive surgery was performed 4–6 weeks after the last cycle of NACT, allowing time for full recovery of the bone marrow and other hematological parameters. Intraoperatively, PCI score was calculated for the 13 regions of the abdomen and pelvis [30]. Additionally, the SB-PCI, adding scores of regions 9, 10, 11, and 12, was separately documented. Peritonectomy and visceral resections were performed with the goal of complete cytoreduction. The extent of completeness of cytoreduction was documented as the CCR (completeness of cytoreduction) score, as described by Sugarbaker et al. [31]. 

Before and after CRS, extended intraperitoneal lavage (EIPL) with 10 L of physiologic saline solution was routinely carried out in all cases undergoing CRS [32]. For HIPEC, 4–5 L of heated saline were instilled in the peritoneal cavity with either mitomycin C 20 mg/m^2^ with cisplatin 40 mg/m^2^ or oxaliplatin 200 mg/m^2^ with 5-fluorouracil 500 mg/m^2^. Few patients received docetaxel 40 mg/m^2^ with cisplatin 40 mg/m^2^ or mitomycin alone or oxaliplatin alone during HIPEC. The drugs used were based on surgeon’s discretion, the treatment policy during that time period, or the patient´s insurance cover. HIPEC was carried out for 40 min and at 43 °C using the open coliseum technique. HIPEC was avoided in case of stormy intra-operative course: excessive blood loss, inability to maintain adequate urine output or mean arterial pressure. Bowel anastomosis was performed after completion of HIPEC.

Postoperative complications occurring from the day of operation until discharge were graded according to the Clavien–Dindo classification [33]. Once patients recovered from CRS and HIPEC, chemotherapy was continued in the adjuvant setting until disease recurrence or progression or unacceptable toxicity. All patients were followed up in the outpatient clinic, with physical examination and tumor marker levels, every 2 months for the first 2 years and then every 6 months thereafter. CT scan was done annually or as clinically indicated. PET-CT or MRI was advised as per deemed necessary by the treating doctor to confirm or rule out recurrence in the case of ambiguity on CT scan findings, especially in patients with normal tumor marker levels. Recurrent disease at the site of anastomosis, regional lymph nodes, and/or peritoneum were included as local recurrence. Recurrence other than these, including non-regional abdominal lymph nodes, was noted as systemic recurrence.

### 4.2. Statistical Analysis

All statistical analyses were performed using the IBM SPSS software version 22 (SPSS Inc., Chicago, DE, USA). Overall survival was calculated from the date of surgery (CRS + HIPEC) to the date of patient’s death or the last follow-up date. Progression-free survival (PFS) was calculated from the date of surgery (CRS + HIPEC) to the date of recurrence (for patients undergoing CC0/1 resection) or date of progression (for patients undergoing CC2/3 resection) or the date of patient’s death. Survival analysis was performed using the Kaplan–Meier method and compared using log-rank test for univariate analysis. Multivariate analysis was carried out by including all factors with a *p* < 0.1 in univariate analysis, using Cox regression model. A *p* < 0.05 was considered statistically significant with 95% confidence interval.

## 5. Conclusions

The results of this study show that, with proper patient selection, with PCI ≤ 12, SB-PCI ≤ 2, complete cytoreduction, and HIPEC, even patients with signet ring subtype of CRC-PM can achieve a median OS of 17.8 to 20.8 months with a 5-year OS of 11.6%. With the ongoing research on the genomic characteristics of the signet ring subtype of CRC, it may be possible to identify activated pathways in these tumors, which can provide us with potential targets for even better treatment of this rare and aggressive disease.

## Figures and Tables

**Figure 1 cancers-12-02536-f001:**
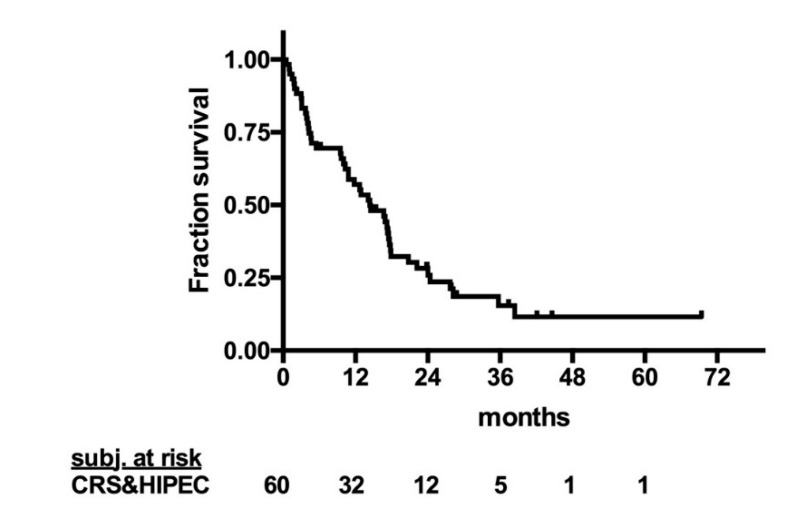
Overall survival of patients with signet ring cell colorectal cancer with peritoneal metastasis treated with CRS and HIPEC. CRS: cytoreductive surgery; HIPEC: hyperthermic intraperitoneal chemotherapy.

**Figure 2 cancers-12-02536-f002:**
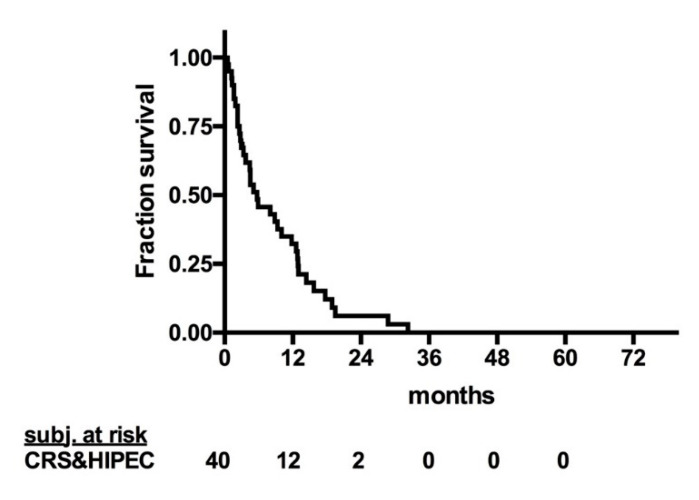
Progression-free survival of patients with signet ring cell colorectal cancer with peritoneal metastasis treated with CRS and HIPEC. CRS: cytoreductive surgery; HIPEC: hyperthermic intraperitoneal chemotherapy

**Figure 3 cancers-12-02536-f003:**
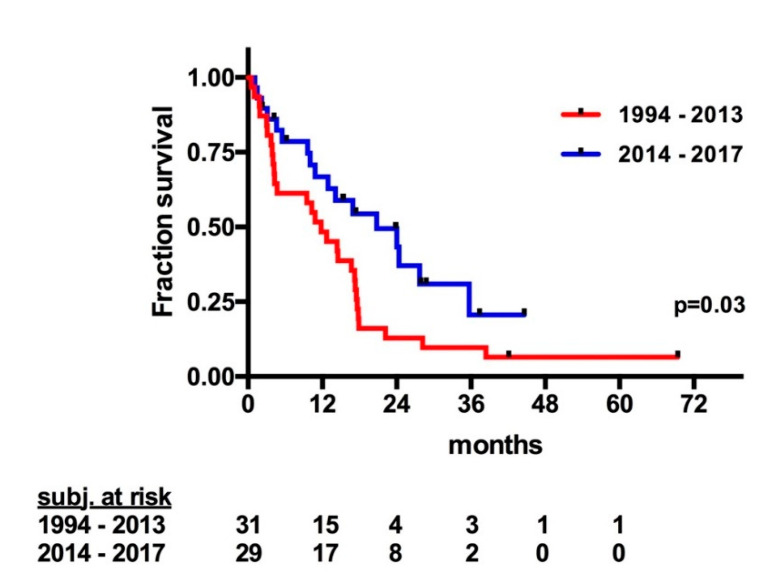
Overall survival of patients with signet ring cell colorectal cancer with peritoneal metastasis treated with CRS and HIPEC regarding the era of treatment. CRS: cytoreductive surgery; HIPEC: hyperthermic intraperitoneal chemotherapy.

**Table 1 cancers-12-02536-t001:** Baseline demographic, clinico-pathological, and treatment characteristics of 60 patients with signet ring subtype of colorectal cancer and peritoneal metastasis treated with CRS ± HIPEC; continuous variables are demonstrated as mean and standard deviation.

Characteristics		Value
Sex (%)	Male	56.7 (34/60)
	Female	43.3 (26/60)
Age (years)		51.4 (SD: 13.9)
Primary Tumor Location (%)	Cecum	20 (12/60)
	Ascending colon	23.3 (14/60)
	Transverse colon	8.3 (5/60)
	Descending colon	3.3 (2/60)
	Sigmoid colon	30 (18/60)
	Recto-sigmoid	15 (9/60)
Appearance of PM (%)	Synchronous	60.3 (35/58)
	Metachronous	39.7 (23/58)
	Not known	2
PCI (Mean)	Total PCI	13.1 (SD: 10.4)
	Small bowel PCI	3.6 (SD: 3.8)
Patients Treated with HIPEC (%)	Yes	75 (45/60)
	No	25 (15/60)
HIPEC Drug Used (%)	MMC + CDDP	60 (27/45)
	Oxaliplatin + 5FU	33.3 (15/45)
	MMC	2.2 (1/45)
	Taxotere + CDDP	2.2 (1/45)
	Oxaliplatin	2.2 (1/45)
Mean Surgical Time (min)		302.6 (SD: 258.8)
Mean Blood Loss (mL)		1457.6 (SD: 1160.5)
Mean No. of Blood Units Transfused		4.4 (SD: 5.7)
Completeness of Cytoreduction (%)	CC0	61.7 (37/60)
	CC1	5 (3/60)
	CC2	13.3 (8/60)
	CC3	20 (12/60)
Clavien–Dindo (%)	Grade 0	50 (30/60)
	Grade I–II	23.3 (14/60)
	Grade III	13.3 (8/60)
	Grade IV	11.7 (7/60)
	Grade V	1.7 (1/60)
Preoperative Chemotherapy (%)	No Chemotherapy	1.7 (1/60)
	NACT	91.7 (55/60)
	NACT without antibody	58.9 (32/60)
	NACT + anti-VEGF antibody	23.6 (13/55)
	NACT + anti EGFR antibody	16.4 (9/55)
	NACT + multiple targeted drugs	1.8 (1/55)
	Biphasic chemotherapy (IV + IP)	5 (3/60)
	Biphasic chemotherapy without antibody	66.6 (2/3)
	Biphasic chemotherapy + anti-VEGF antibody	0 (0/3)
	Biphasic chemotherapy + anti EGFR antibody	33.3 (1/3)
	No details	1.7 (1/60)
Postoperative Chemotherapy (%)	No chemotherapy	1.7 (1/60)
	Systemic chemotherapy	63.3 (38/60)
	Systemic chemotherapy without antibody	50 (19/38)
	Systemic chemotherapy + anti-VEGF antibody	26.3 (10/38)
	Systemic chemotherapy + anti EGFR antibody	18.4 (7/38)
	Systemic chemotherapy + multiple targeted drugs	5.2 (2/38)
	Biphasic chemotherapy (IV + IP)	1.7 (1/60)
	No details	33.3 (20/60)

CRS: cytoreductive surgery; PM: peritoneal metastasis; PCI: peritoneal carcinomatosis index; SD: Standard Deviation; No: number; HIPEC: hyperthermic intraperitoneal chemotherapy; MMC: mitomycin C; CDDP: cisplatin; 5FU: 5-fluorouracil; NACT: neoadjuvant chemotherapy; VEGF: vascular endothelial growth factor; EGFR: epidermal growth factor receptor; IV: intravenous; IP: intraperitoneal; CC: completeness of cytoreduction.

**Table 2 cancers-12-02536-t002:** Uni- and multivariate analyses of factors affecting survival among signet ring subtype of colorectal cancer patients with peritoneal metastasis after CRS ± HIPEC. Overall survival is illustrated as median with 25% and 75% quartiles.

Variable	Category	No.	Median Overall Survival (Months)	Univariate Analysis (*p* Value)	Multivariate Analysis		
					HR	95% CI	*p* Value
Gender	Female Male	26 34	24.4 (10.1–38.7) 11.8 (8.6– 5.1)	0.004	2.14	0.94–4.89	0.071
Appearance of PM	Synchronous Metachronous	35 23	14.4 (11.9–16.9) 16.7 (7.6–25.7)	0.960			
HIPEC	HIPEC No HIPEC	45 15	17.3 (16.2–18.4) 5.5 (0–16.1)	0.007	2.20	1.01–4.81	0.049
Complications	No Yes	30 30	17.6 (3.5–31.70) 10.8 (0–23)	0.006			
CCR	CC0-1 CC2-3	40 20	17.6 (16.7– 8.5) 4.7 (0–12)	<0.001	2.26	0.94–5.42	0.068
Intraoperative Cytology	Negative Positive	29 22	17.6 (13.1–22.1) 9.6 (1.8–17.3)	0.017	1.78	0.86–3.71	0.122
Age	≤60 years >60 years	40 20	12.9 (9.0–16.8) 17.5 (2.6–32.4)	0.219			
Nodal status	N0 N+	15 45	16.9 (12.6–21.2) 12.9 (7.7–18.1)	0.690			
Side of Primary	Right Left	31 29	14.1 (9.8–18.4) 16.7 (9.6–23.8)	0.829			
PCI	≤12 >12	32 27	17.8 (10.1–25.5) 10.0 (8.4–11.5)	0.004	0.27	0.07–1.10	0.068
Small Bowel PCI	≤2 >2	31 25	20.8 (13.1–28.5) 3.5 (5.8–15.5)	<0.001	6.51	1.65–25.74	0.008
Blood loss	≤1000 mL >1000 mL	22 35	17.2 (7.7–26.7) 14.5 (8.9–20.1)	0.398			
BT	No Yes	20 28	17.3 (13.1–1.5) 14.5 (7.1–22.0)	0.295			

HR: hazard ratio; CI: confidence interval; PM: peritoneal metastasis; HIPEC: hyperthermic intraperitoneal chemotherapy; CRS: Cytoreductive Surgery; PCI: peritoneal carcinomatosis index; CC: completeness of cytoreduction; BT: blood transfusion.

**Table 3 cancers-12-02536-t003:** Study demographics and patient outcomes of published studies about CRS and HIPEC in patients with signet ring cell subtype of colorectal cancer.

References	Total Number of Patients	Patients with Signet Ring Cell	Median OS	5-Year OS	Mean PCI	CC0/CC1
Verwaal et al. 2004 [23]	102	15	13	NR	NR	49% *
Chua et al. 2009 [24]	33	15	13	NR	NR	46.7%
van Sweringen et al. 2012 [25]	36	5	7.2 ^#^	NR	15 ^#^	83% *
Winer et al. 2014 [26]	67	30	12.2	7%	11.5	77%
van Oudheusden et al. 2014 [27]	351	20	14.1	NR	NR	87.5%

CRS: cytoreductive surgery; HIPEC: hyperthermic intraperitoneal chemotherapy; OS: overall survival; CC0/CC1: complete macroscopic cytoreduction; NR: not reported; * reported for full cohort only; # reported for 11 patients with signet ring cell subtype of different origins.

## Data Availability

The data that support the findings of this study are available on request from the corresponding author. The data are not publicly available due to privacy or ethical restrictions.

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
