# Peer review of "Retrospective Analysis of Patients with Signet Ring Subtype of Colorectal Cancer with Peritoneal Metastasis Treated with CRS & HIPEC"

_cancers, 2020, doi:10.3390/cancers12092536_

Round 1

Reviewer 1 Report

Congratulations, the authors were able to collect very large amounts (60) with colon signet cell carcinoma. Data were analyzed very carefully, good statistical analysis. Nevertheless, there are shortcomings in this article that the authors keep silent about. Well first, it’s a retrospective data analysis. Secondly, it is a very long period of analysis of data - 23 years. During that period, attitudes towards HIPEC changed, the assessment of the effectiveness of medicines used for the HIPEC procedure changed, HIPEC machines improved, and the possibilities for data recording changed. Ignoring this diminishes the power and curiosity of the article. I think the authors could mention these shortcomings in the interlude, or divide the data of the patients in question into two periods. This would provide an opportunity to assess modern standards for the application of HIPEC and CRS.

Author Response

Reviewer #1:

Congratulations, the authors were able to collect very large amounts (60) with colon signet cell carcinoma. Data were analyzed very carefully, good statistical analysis. Nevertheless, there are shortcomings in this article that the authors keep silent about. Well first, it’s a retrospective data analysis. Secondly, it is a very long period of analysis of data - 23 years. During that period, attitudes towards HIPEC changed, the assessment of the effectiveness of medicines used for the HIPEC procedure changed, HIPEC machines improved, and the possibilities for data recording changed. Ignoring this diminishes the power and curiosity of the article. I think the authors could mention these shortcomings in the interlude, or divide the data of the patients in question into two periods. This would provide an opportunity to assess modern standards for the application of HIPEC and CRS.

RE: Thank you for your valuable comment. We added these shortcomings to the paragraph of study limitations. We performed an analysis comparing the overall survival of patients treated between 1994 – 2013 with patients treated between 2014 – 2017 (selection of year according to numbers per group; 31 vs. 29 patients). There was a significant difference in median overall survival (11.8 vs. 20.8 months; p=0.03). We added this Figure to the Results section, and mentioned it in the Discussion section.

Reviewer 2 Report

This retrospective analysis of a rare subgroup of colorectal cancer patients with peritoneal metastases treated with CRS +/- HIPEC is a valuable contribution based on the relatively large number of patients included. The authors have successfully uncovered a number of prognostic factors. They do mention some of the limitations of their study, and maybe they should also discuss the strong variation in chemotherapeutic HIPEC agents used as a limitation. I was also wondering how come that they had 60 out of 320 patients from this rare subtype, when they mention that normally only about 1% of colorectal patients are of this subtype. But overall a valuable study.

Author Response

Reviewer #2:

This retrospective analysis of a rare subgroup of colorectal cancer patients with peritoneal metastases treated with CRS +/- HIPEC is a valuable contribution based on the relatively large number of patients included. The authors have successfully uncovered a number of prognostic factors. They do mention some of the limitations of their study, and maybe they should also discuss the strong variation in chemotherapeutic HIPEC agents used as a limitation. I was also wondering how come that they had 60 out of 320 patients from this rare subtype, when they mention that normally only about 1% of colorectal patients are of this subtype. But overall a valuable study.

RE: Thank you for your important comment. We added this aspect to the paragraph of study limitations. The incidence of signet ring cell subtype of CRC of 18.8% (60/320) in our study is higher compared to other studies focusing on patients treated with CRS. Kozman et al. reported an incidence of 6.9%, Ihemelandu et al. of 6.8%, and Cashin et al. of 11%, respectively [1-3]. It seems that signet ring cell subtype is higher represented in the metastasized subgroup due to its aggressiveness, but we are not able to explain the different incidence of our Japanese cohort compared to cohorts from Australia, U.S.A. or Sweden.

References

  1. Cashin PH, Graf W, Nygren P, Mahteme H. Patient selection for cytoreductive surgery in colorectal peritoneal carcinomatosis using serum tumor markers: an observational cohort study. Ann Surg. 2012;256(6):1078-83.
  2. Ihemelandu C, Sugarbaker PH. Management for Peritoneal Metastasis of Colonic Origin: Role of Cytoreductive Surgery and Perioperative Intraperitoneal Chemotherapy: A Single Institution's Experience During Two Decades. Ann Surg Oncol. 2017;24(4):898-905.
  3. Kozman MA, Fisher OM, Rebolledo BJ, Parikh R, Valle SJ, Arrowaili A, et al. CEA to peritoneal carcinomatosis index (PCI) ratio is prognostic in patients with colorectal cancer peritoneal carcinomatosis undergoing cytoreduction surgery and intraperitoneal chemotherapy: A retrospective cohort study. J Surg Oncol. 2018;117(4):725-36.